# Phytochemicals and Traditional Use of Two Southernmost Chilean Berry Fruits: Murta (*Ugni molinae* Turcz) and Calafate (*Berberis buxifolia* Lam.)

**DOI:** 10.3390/foods9010054

**Published:** 2020-01-06

**Authors:** Carolina Fredes, Alejandra Parada, Jaime Salinas, Paz Robert

**Affiliations:** 1Departamento Ciencias de la Salud, Carrera de Nutrición y Dietética, Facultad de Medicina, Pontificia Universidad Católica de Chile, Santiago 7820436, Chile; cpfredes@uc.cl (C.F.); acparada@uc.cl (A.P.); 2Instituto Forestal, Sede Coyhaique, Coyhaique 5951840, Chile; jsalinas@infor.cl; 3Departamento Ciencia de los Alimentos y Tecnología Química, Facultad de Ciencias Químicas y Farmacéuticas, Universidad de Chile, Santiago 8380492, Chile

**Keywords:** anti-inflammatory, antimicrobial, antioxidant, anthocyanins, medicinal foods, nutraceuticals

## Abstract

Murta and calafate have been traditionally used by indigenous and rural peoples of Chile. Research on murta and calafate has gained interest due to their attractive sensory properties as well as a global trend in finding new fruits with potential health benefits. The objective of this review was to summarize the potential use of murta and calafate as sources of nutraceuticals regarding both the traditional and the up-to-date scientific knowledge. A search of historical documents recorded in the Digital National Library as well as scientific articles in the Web of Science database were performed using combinations of keywords with the botanical nomenclature. Peer-reviewed scientific articles did meet the inclusion criteria (*n* = 38) were classified in phytochemicals (21 papers) and biological activity (17 papers). Murta and calafate are high oxygen radical absorbance capacity (ORAC)-value fruits and promising sources of natural antioxidants, antimicrobial, and vasodilator compounds with nutraceutical potential. The bioactivity of anthocyanin metabolites in murta and calafate must continue to be studied in order to achieve adequate information on the biological activity and health-promoting effects derived for the consumption of murta and calafate fruit.

## 1. Introduction

Historically, indigenous peoples of Chile have had a deep relationship with nature; flora native to their territories has been used for various purposes, such as food, fuel, religious ceremonies, decoration, dyeing, and medicine [1]. In this context, the Mapuche (“people of the earth” in the Mapuzugun language) hold a vast, rich body of knowledge about flora that has been learned and transmitted within the culture throughout space and time [2,3]. Moreover, southernmost communities, such as the Aónikenk (“people of the south” in the Aónikoaish language) and the Yámana (“human being” in the Yahgan language), collected several edible plant roots, wild fruits, and seeds to survive in extremally harsh conditions [4]. In these particular areas, two small berry-type fruits known as murta, murtilla, or uñi (*Ugni molinae* Turcz, *Myrtaceae*) and calafate (*Berberis buxifolia* Lam., *Berberidaceae*) grow in the wild of the Patagonia. Murta is an evergreen bush that naturally grows in Chile from south of Talca (35° SL) to the Palena River (44° SL) (Figure 1), forming part of the deciduous forests of *Nothofagus* as well as the southern evergreen forests [5]. In these habitats, murta grows alongside other edible Chilean fruit plants such as peumo (*Cryptocarya alba*), boldo (*Peumus boldus*), keule (*Gomortega queule*), avellano or gevuin (*Gevuina avellana*), diverse michay species (*Berberis darwinii*, *B. serrata*, *B. dentata*), litre (*Lithraea caustica*), pitra (*Myrceugenia planipes*), and luma (*Amomyrtus luma*) [1]. Calafate is an evergreen spiny bush which naturally grows in Chile from Curicó (35° SL) to the Cape Horn Archipelago (56° SL) [6]. Nevertheless, it is most abundant from Valdivia (40° SL) to the Strait of Magellan (54° SL) (Figure 1). 

During the last decade, interest in studying these berry-type fruits has increased, due to their attractive sensory properties as well as a global trend in finding new fruits with potential health benefits. Berry-type fruits are well known for their high polyphenol, and especially anthocyanin content [7]. Specifically, wild berry-type fruits stand out over their cultivated counterparts in terms of polyphenol content [8]. Furthermore, the extreme weather conditions that predominate during southern Chilean summers (a high-temperature oscillation) may favor the plant biosynthesis of anthocyanins [9]. Both murta and calafate fruits can be sources of anthocyanins with nutraceutical potential. Lee [10] defined Nutraceuticals as foods or part of foods that provide both health benefits to reduce the risk of chronic diseases and basic nutrition. Therefore, this review summarizes the potential use of murta and calafate as sources of nutraceuticals regarding both the traditional and the up-to-date scientific knowledge. This review would like to contribute to the development of both new research and indigenous peoples, in line with the 2030 agenda for sustainable development goals of the United Nations with its promise to “leave no one behind”.

## 2. Traditional Knowledge around Murta and Calafate 

The search strategy of historical documents was focused in revising manuscripts written by recognized naturalists of the Chilean flora such as Juan Ignacio Molina (1740–1829) [11], Claude Gay (1800–1873) [12,13], Charles Darwin (1809–1882) [4], and Ernesto Wilhelm de Mosbach (1882–1963) [1] available in Digital National Library database (www.memoriachilena.cl). At the same time, ethnobotany and ethnopharmacology books were also suitable publications that were extensively reviewed. 

### 2.1. Murta and Calafate in the Mapuche and Rural Culture

The Mapuche are the most prominent indigenous peoples in Chile, due to both their social and demographic weight and cultural identity. Mapuche have historically settled between the Itata (36° SL) and Toltén (38° SL) Rivers (Figure 1) in Chile [2]. In this sense, Figure 1 contrasts the wild distribution of murta and calafate and the historically territory occupied by the Mapuche people.

The first records regarding the traditional uses of murta fruit (Figure 2a) by the Mapuche indicate the preparation of a sweet stomach wine that was an appetite stimulant, and “whose aroma was appreciated as the most delicate Muscat” [11]. In this sense, the aromatic properties of murta fruits were early recognized; Gay [13] indicated that country people eat murta fruits with a great pleasure and prepared pleasant and aromatic confections. Nowadays people from the countryside consume murta as dried fruits and in the preparation of jams [14].

Mapuche used the word *maki* to name black fruits [1]. In this sense, calafate (Figure 2b) were appreciated as fruit rich in pigments where the root and bark of calafate were also used to obtain purple and yellow dyes [14]. 

The Mapuche tradition of using plants for medicinal purposes was recorded in the archives of the several settlers and naturalists who, in turn, enriched them with the contribution of medicinal plants from Europe and other regions [15]. Since 2008, the traditional use of medicinal plants has been recognized by the Chilean Health Ministry [16]. The traditional use of murta includes making a leaf infusion to treat urinary and throat infections, while the fruit is known for its astringent power [14,17]. In the case of calafate, traditional uses include using its roots to control fevers, as an anti-inflammatory, and to ease stomach pains, indigestion, and colitis [14,17].

### 2.2. Calafate in the Extreme South Communities of Chile 

Historical records on the use of native flora as food and medicine by extreme south communities are more limited than the information available about the indigenous peoples of the central and southern zone of Chile. Among the communities of the extreme south Chile, the Aónikenk were an exclusively cultural entity defined by a particular tradition, language, and lifestyle [18], who occupied the territory located between the Santa Cruz River (50° SL) and the Strait of Magellan (54° SL) (Figure 1). The Aónikenk had a significant collecting tradition in which not only women participated but also involved men, children, and occasionally elderly people [18]. Calafate as well as other native fruits like zarzaparrilla (*Ribes magellanicum*), chaura (*Pernettya mucronata*), murtilla (*Empetrum rubrum*), and strawberry (*Rubus geoides*) were consumed mainly as fresh fruits during the harvest season (November to March) [4]. Furthermore, Aónikenk women occasionally painted their faces with calafate juice, believing it would whiten the skin tone [18].

The Yámana were the southernmost ethnic group in the world who lived in the territory of the Cape Horn Archipelago (56° SL) in the south of Tierra del Fuego Island and the Beagle Channel (Figure 1) [18]. Yámana were maritime hunter-gatherers who spent much of their lives in their anan (tree bark canoe). Men collected firewood, fruits, and vegetables [19]. A Yámana legend says that whoever tastes the bittersweet flavor of calafate will return to the extreme south territory [20].

## 3. Scientific Knowledge around Murta and Calafate 

The search strategy of scientific literature is detailed in the flow diagram of Figure 3. 

Relevant publications were identified through an initial search in Web of Science database using the current botanical nomenclature in the title. The time interval for obtaining the records was between 1975 and 2020. Since 1994, the search results identified a higher number of documents on murta (58 records) than on calafate (31 records). Each abstract was revised and publications associated to plant sciences, environmental sciences and ecology that did not report any information about nutritional and phytochemical composition of murta and calafate (36 records) were excluded. At the same time, publications that did not contain original data (four records) were also excluded. Suitable publications were extensively reviewed and scientific articles contained the same data as other studies (11 papers) were excluded. Papers did meet the inclusion criteria were classified in phytochemicals (21 papers) and biological activity (17 papers) in terms to analize the the up-to-date scientific evidence of murta and calafate as sources of nutraceuticals. 

### 3.1. Nutritional Content of Murta and Calafate Fruits

Murta fruit has a soluble solid content (SS) around 19 °Brix, a titratable acid around 8 meq Na OH/100 g, and a pH around 4.3 [21]. The sensory properties of murta fruit include the sweetness of a strawberry, the pungency of a guava, and the texture of a dried blueberry [22]. Furthermore, volatile compounds identified in murta fruit aroma have been described as fruity, sweet, and floral; ethyl hexanoate and 4-methoxy-2,5-dimethyl-furan-3-one are the most potent compounds found in murta [23]. The proximate analysis of murta fruit shows a content in moisture of 76.9%, protein of 1.2 g/100 g, fat of 0.9 g/100 g, crude fibre of 2.5 g/100 g, ash of 0.5 g/100 g and carbohydrates (by difference) of 18.5 g/100g expressed on fresh weight (FW) [24]. Furthermore, murta has a content in ascorbic acid of 210 mg/100 g [25]. 

Calafate fruit has a SS content between 25–31 °Brix and a titratable acid around 2 g malic acid/100 g (15 meq NaOH/100 g) [26]. Calafate fruit has a higher SS content than murta, and the highest SS content in comparison to other berries such as maqui (19 °Brix), pomegranate (15 °Brix), blueberry (14 °Brix), blackberry (13 °Brix), red raspberry (11 °Brix), and strawberry (9 °Brix) [27]. The proximate composition, sugars, and ascorbic acid of calafate fruit shows a content in moisture of 75%, protein of 2.6 g/100 g, ether extract of 0.18 g/100 g, crude fibre of 7.0 g/100 g, ash of 0.9 g/100 g, glucose of 3.0 g/100 g, fructose of 0.8 g/100 g and ascorbic acid of 74.0 mg/100 g expressed on FW [28]. 

### 3.2. Phytochemicals in Murta and Calafate Fruits

Phytochemicals (i.e., phenolic compounds, terpenoids, alkaloids, and nitrogen-containing compounds) can be defined as non-nutrient chemicals found in plants that demonstrate biological activity against chronic diseases [29,30]. Furthermore, several phytochemicals such as phenolic compounds and terpenoids have been used throughout human history as condiments and pigments [31]. 

#### 3.2.1. Anthocyanins in Murta and Calafate Fruit

Anthocyanins are the common coloring compounds found in a large number of plants and are responsible for the purple, red, blue, and orange colors of berry-type fruits [32]. Chemically, anthocyanins are phenolic compounds belonging to the flavonoid class, with two benzene rings joined by a linear three-carbon chain, possessing the C6–C3–C6 basic skeleton [33]. Anthocyanins are formed by the modification of anthocyanidins by glycosyl and aromatic or aliphatic acyl moieties [34]. The basic structure of an anthocyanidin is a flavonoid ion (2-phenylbenzopyrilium) that can vary based on the different positions of hydroxyl groups (OH) or methoxyls (OCH_3_) (Figure 4a). Six anthocyanidins (cyanidin, delphinidin, pelargonidin, peonidin, malvidin, and petunidin) are the most common in plants [33,35] and are present in berry-type fruits [32,36].

The use of LC-MS as an analytical chemistry technique has allowed for the identification and/or tentative identification of the anthocyanin profile of murta and calafate. In a first report, two anthocyanins (cyanidin-3-*O*-glucoside and peonidin-3-*O*-glucoside) were identified in murta whereas 18 anthocyanins (3-*O*-monoglycosylated and 3,5-*O*-diglycosylated delphinidins, cyanidins, petunidins, peonidins, and malvidins) were most abundant in calafate [28,37,38]. As can be expected by the wide variability of colors observed in murta ecotypes, subsequent studies [39] have described a greater number of anthocyanins where 3-*O*-glucosides of delphinidin, petunidin, and malvidin have been also identified in murta fruit. Based on these findings, murta (Figure 4a) and calafate (Figure 4b) could be used as promising sources of a wide variety of anthocyanins.

#### 3.2.2. Phenolic Acids and Other Flavonoids in Murta and Calafate Fruit

The identification of other phenolic compounds in murta and calafate indicate the presence of phenolic acids (gallic acid, benzoic acid, p-coumaric acid, and hydrocaffeic acid), flavan-3-ols (epicatechin) and flavonols (quercetin, rutin, luteolin, kaempferol, and myricetin) in murta fruits [39,40]. On the other hand, 20 hydroxycinnamic acids and flavonols such as hyperoside, isoquercitrin, quercetin, rutin, myricetin, and isorhamnetin are more abundant in calafate fruits [39,41,42]. Furthermore, among flavonoid subclasses, murta shows higher flavonol and flava-3-ols content (0.29 and 0.27 µmoL/g FW respectively) than calafate (0.16 and 0.24 µmoL/g FW respectively) whereas calafate has a higher anthocyanin content (17.8 µmoL/g FW) than murta (0.21 µmoL/g FW) associated with the deep-blue color of the calafate fruit [28].

The existing research on murta and calafate show that phenolic compounds are the main phytochemicals studied in these berry-type fruits.

### 3.3. Phytochemical Changes in Murta and Calafate

The main factors that affect the polyphenol content in plants are genotype, environment, stage at fruit harvest, and storage and processing [15]. When wild plants are studied, domestication emerges as a key factor affecting polyphenol content due to changes in the allocation of nutrients within the plant (domestication syndrome) that occur during the continuous process of selection [43]. 

#### 3.3.1. Genotype vs. Environment

Phytochemical changes in murta have been studied in its leaves and stems and its fruits. The total amount of flavonols (rutin, kaempferol, and quercetin) in wild murta plant (leaves) is significantly higher (~20%) than in cultivated murta plants, showing the domestication effect [44]. Results on the flavonol content in murta fruit have been inconsistent. In one study, no significant differences between wild and cultivated murta fruits were found [44,45]. Conversely, in another study by Augusto [40], the total phenolic compounds in the fruits of wild and selected murta (14-4) genotype were 19.4 and 40.3 mg GAE/g in dry weight, respectively. Consistent with the results of phenolic compound content, the antioxidant capacity (AC) of the wild murta genotype was lower than the selected murta (14-4) genotype for both DPPH (76.5 and 134.4 mu moL TEAC/g) and ABTS (157.0 and 294.0 mu moL TEAC/g) tests [40]. These results show that phenolic content and AC in murta fruits did not decrease as a result of the domestication process. 

In addition to differences in polyphenol content among different murta genotypes, the effect of the environment may cause the same variety to present differences depending on the harvest year. In this context, Alfaro et al. [46] indicate that genotype and growing season have a significant effect on the polyphenol content and AC of murta fruits; the lowest polyphenol content (283 ± 72 mg GAE/100 g DW) was obtained for the 14-4 genotype in 2008, and the highest value (2152 ± 290 mg GAE/100 g DW) was observed for the variety South Pearl-INIA in 2007. Furthermore, a canonical discriminant analysis of seasonal differences showed that the South Pearl-INIA variety had the greatest variation in polyphenol content in relation to the other genotypes studied [46]. The same authors indicate that rainfall and frosts were the most relevant climate factors that may explain the seasonal variation of total polyphenol content of murta fruits [46].

In the case of calafate, the effect of genotype and environment on phytochemical changes is scarce. Mariangel et al. [47] indicated that the phenolic composition of calafate fruit can vary according to the geographical area of fruit collection. However, it is difficult to determine if these differences are attributed to the different genotypes evaluated and/or to the effect of the environment. On the other hand, Arena et al. [48] demonstrated that calafate fruits from field conditions with different light intensities (i.e., 100%, 57%, 24% of natural irradiance) show significant differences in anthocyanin content. Fruits grown under high light intensity (299.7 mg/100 g FW) had an anthocyanin content 2.9 times higher than fruits grown under medium light intensity (103.8 mg/100 g FW) [48]. The authors associated a higher anthocyanin content with a higher photosynthetic rate and a concomitant increase in SS and sugar content measured under the same conditions [48].

#### 3.3.2. Stage at Fruit Harvest

Berry-type fruits harvested at different maturity stages present different chemical characteristics as well as phenolic compound profiles; in immature fruits proanthocyanidins are predominant while in mature fruits anthocyanins predominate [49,50,51]. From an ecological point of view, plants with immature fruits—without viable seeds—have high phenolic and proanthocyanidin contents that act as deterrent compounds to prevent their consumption by insects and herbivores [52]. In contrast, ripe fruits have attractive colors (anthocyanins and other pigments) and a sweet taste to stimulate their consumption by herbivores and the consequent dispersion of seeds [52].

Phytochemical changes during fruit development and ripening have been studied in calafate fruits as well as other berry-type fruits [49,50,51,53]. Early studies attempted to understand the main changes in phenolic compounds and AC during the process of fruit maturation and ripening in order to establish the best maturity index for calafate fruit according to phenolic compound content [26,54]. According to this study, for calafate fruits collected in March, the maximum anthocyanin content (761 mg/100 g FW) coincided with the highest accumulation of SS (25 °Brix) 126 days after flowering [54].

#### 3.3.3. Storage and Fruit Processing

In fruits, anthocyanins are mainly found in the epicarp (peel) [55] where they are stored in the cell vacuole and remain stable in intact fruits [56]. During fruit processing, the plant cell loses its compartmentalization and anthocyanins are exposed to the action of enzymes (polyphenol oxidase), other fruit components, and environmental conditions that favor their degradation [57]. Several factors such as temperature, pH, light, oxygen, ascorbic acid, metal ions, sugars, and enzymes may affect the stability of anthocyanins during processing and storage [58]. The mechanism of anthocyanin degradation has been studied in some plant species such as roselle (*Hibiscus sabdariffa*); a metal-catalyzed oxidation followed by condensation (brown polymer) and a deglycosylation followed by scission (phloroglucinaldehyde and phenolic acids) were identified as two pathways of anthocyanin degradation [59].

Fleshy fruits have high moisture content, thereby they are classified as highly perishable commodities [60]. The dehydration of murta and calafate fruits emerges as an attractive conservation alternative given also the high seasonality of their production, allowing for the commercialization of value-added dehydrated fruit products throughout the year. Different drying processes for murta, such as freeze-drying [61,62], convective drying, combined infrared-convective drying [63], vacuum drying [62,64,65], atmospheric drying [62,64], infrared-radiation [62], and sun-drying [62] have been evaluated in order to preserve the polyphenols, the AC, and/or the microstructure of fresh murta fruit. Lopez et al. [62] reported that freeze-drying showed the highest retention of total flavonoids as well as anthocyanins and the least damage of murta microstructure. No studies on the drying of calafate fruit have been published. 

The understanding of the phytochemical changes in murta and calafate facilitates the design and development of formulations and the evaluation of the efficacy of new nutraceutical products.

### 3.4. Validation of Traditional Use and New Insights in the Research of Murta and Calafate

Different methods of biological activity have been used to validate the traditional uses of murta and calafate as well as to examine some potential uses of calafate shoots and fruits (Table 1). 

Some studies identified phytochemicals such as triterpenoids, phenolic compounds, and isoquinoline alkaloids. However, in many cases it is difficult to correlate a specific compound to a specific biological effect. In addition, some studies lacked detailed information about the genotype and time of harvest of raw materials. This information is particularly important because, as was discussed previously, genotype, environment, and time of harvest have been shown to affect the phytochemistry of murta and calafate. 

#### 3.4.1. Antioxidant Capacity

Based on a traditional use of murta leaf that suggests antioxidant, anti-inflammatory, and antimicrobial activity, an early study of murta leaf AC (ORAC method) in vivo was evaluated in the plasma of healthy volunteers before and after the ingestion of a murta leaf infusion (1%) twice a day for three days [66]. The results indicated a significant increase (from 2.258 to 3.108 μM TE/L) in the AC of the volunteers’ plasma. Later, murta leaf AC was attributed to the presence of polyphenols such as phenolic acids, hydrolyzable tannins, flavanols (epicatechin), and flavonols (myricetin, quercetin) [67]. In another study, Albrecht et al. [82] examined the beneficial effect of calafate fruit on oxidative stress induced by chloramphenicol. Calafate-fruit aqueous extracts were shown to reduce oxidative stress caused by chloramphenicol in human blood cells by significantly diminishing reactive oxygen species (ROS). In parallel with the decrease of ROS, the fruit extract protected the viability of leukocytes [82]. 

#### 3.4.2. Anti-Inflammatory Activity 

To study the biological activity of murta-leaf triterpenoids, the topical anti-inflammatory activity of alphitolic, asiatic, and corosolic acids isolated from murta leaf were evaluated in vivo in a mouse ear model; inflammation was induced with either arachidonic acid or 12-*O*-tetradecanoylphorbol-13 acetate [69]. Only corosolic acid was active in the arachidonic acid-induced inflammation assay, with similar potency to nimesulide, while the three triterpene acids together inhibited 12-*O*-tetradecanoylphorbol-13 acetate-induced inflammation with potencies comparable to that of indomethacin [69]. Furthermore, Goity et al. [70] and Arancibia-Radich et al. [71] indicated that the differences in the anti-inflammatory activity of murta leaf is associated to the different quantitative composition of phenolic compounds and triterpenoids. 

The anti-inflammatory potential of calafate fruit has been studied by Reyes-Farias et al. [83] where aqueous extracts were able to modulate the proinflammatory state generated by the interaction between adipocytes and macrophages in vitro.

#### 3.4.3. Antimicrobial Activity

The antimicrobial activity of murta leaf extract against clinically important microorganisms with antibiotic resistance (*Staphylococcus aureus*, *Enterobacter aerogenes*, *Pseudomonas aeruginosa*, and *Candida albicans*) in vitro has been shown by Avello et al. [72] and Shen et al. [73]. Furthermore, Shene et al. [74] described the antimicrobial activity of murta leaf in human gut bacteria and Di Castillo et al. [75] showed the antimicrobial activity of murta leaf against *Escherichia coli* and *Listeria monocytogenes*. 

Junqueira-Gonzalves et al. [39] studied the antibacterial activity (*E. coli* (ATCC 25922) and *Salmonella typhi* (ATCC 14028)) of ethanolic and acidic methanolic extracts. A methanolic murta fruit extract (100 μL) was equivalent to the activity of all of the antibiotics (tetracycline, clotrimazole, gentamicin, amikacin, ceftriaxone, cefuroxim, cefotaxim, ampicillin, ciprofloxacin, and ampicillin/sulbactam) tested in the case of *S. typhi*. However, in the case of *E. coli*, 100 μL of the extract was equivalent to the activity of tetracycline, amikacin, cefuroxim, cefotaxim, ampicillin, and ciprofloxacin. 

The antimicrobial activity of calafate roots and shoots (stem and leaves) against Gram-positive bacteria (*Staphylococcus aureus*, *Bacillus cereus*, *Staphylococcus epidermidis*, and *Bacillus subtilis*) has been associated with the presence of isoquinoline alkaloids [78,81]. Calafate root had the highest alkaloid yield and berberine was the main alkaloid identified [79]. 

#### 3.4.4. Analgesic Activity

Delporte et al. [68] studied the analgesic activity of dichloromethane, ethyl acetate, and methanol extracts from murta leaves on acute pain in mice. Murta-leaf extracts produced antinociception in chemical and thermal pain models through a mechanism partially linked to either lipooxygenase and/or cyclooxygenase via the arachidonic acid cascade and/or opioid receptors. Flavonoids and triterpenoids were associated with the antinociceptive activity [68].

#### 3.4.5. New Insights in the Research of Murta and Calafate

In order to explore in the potential beneficial effect of murta and calafate fruit on the management of cardiovascular disease, Jofre et al. [76] and Calfío and Huidobro-Toro [77] studied the antioxidant and vasodilator activity of murta and calafate fruit in rat models. Dose-dependent vasodilator activity in the presence of endothelium was shown in aortic rings. Its hypotensive mechanism is partially mediated by nitric oxide synthase/guanylate cyclase and large-conductance calcium-dependent potassium channels [76]. Similarly, vascular responses of main glycosylated anthocyanins found in calafate fruit were endothelium-dependent and mediated by NO production [77]. Nevertheless, the authors propose that the anthocyanin-induced vasodilation is not due to an antioxidant mechanism [77]. Furthermore, Furrianca et al. [80] showed that a calafate-root ethanolic extract had hypoglycemic effects, stimulating glucose uptake in non-resistant and insulin-resistant liver (HepG2) cells by activating AMPK protein.

### 3.5. Potential Health Benefits Associated with Murta and Calafate Fruit Consumption

The high consumption of anthocyanin-rich foods has been associated with several health benefits in humans [84,85,86,87]. Internationally, these potential anti-inflammatory, antioxidant, hypoglycaemic, and cardioprotective health benefits found in berry-type fruits are used as a strategy to promote consumption. Antioxidant capacity measured in vitro is commonly determined in studies of phenolic profiling [9,88] however the in vitro method of measuring total antioxidant capacity is questionable due to it having almost zero relevance for human (animal) physiology. Nevertheless, the ORAC value as a way to determine AC in vitro is considered a quality parameter in the international market of berry-type agri-foods [88]. Speisky et al. [89] reported 27 fruit species grown in Chile, where the total phenolic content of murta (863 mg GAE/100g FW) and calafate (1201 mg GAE/100g FW) were higher than well-known polyphenol-rich fruits such as blackberry (671 mg GAE/100g FW) and blueberry (529 mg GAE/100g FW). Murta and calafate were grouped among the highest ORAC (10,000–25,000 µmoL TE/100 g FW) fruits where calafate had the highest ORAC (25,662 µmoL TE/100 g FW) value; 2.8-fold higher than blackberry and 2.9-fold higher than blueberry [89]. 

As was mentioned previously, polyphenols are susceptible to degradation by heat, oxygen, and changes in pH, among others that may occur not only during product storage, but also into the gastrointestinal (GI) tract [57]. For example, it is well known that anthocyanins are unstable at high pH, and the shift from the acidic pH (pH 2) of the stomach to the almost neutral pH of the duodenum (pH 6) may be responsible for their specific hydrolysis and/or degradation [90,91,92]. Bioaccessibility, defined as the amount of compounds that are released from the food matrix after digestion [90] is measured to determine the impact of the food matrix on the protection and/or release of bioactive compounds as well as the stability of bioactive compounds during GI digestion. Ah-Hen et al. [93] compared the bioaccessibility of murta fruit and juice during an in vitro GI digestion process showing that juice as food matrix released bioactive compounds earlier in the gastric stage, while murta fruit released bioactive compounds in the small intestine. However, both murta fruit and juice achieved a high bioaccessibility index of polyphenols (70%) after being digested by the small intestine [93]. 

According to anthocyanin metabolism, its degradation is a result of chemical instability and the impact of bacterial catabolism, resulting in a number of circulating phenolic metabolites [94]. Along this line, Bustamante et al. [95] performed a pharmacokinetic study of phenolic compounds in gerbil plasma after the consumption of calafate, where the amount of 16 phenolic acids increased 4–8 h post-intake. Although all catabolites were found in concentration peaks between 0.1 and 1 mu M, no parental anthocyanins were detected [95]. Currently, it is postulated that anthocyanin bioactivity in vivo results from lesser studied, though more bioavailable, phenolic metabolites [96,97] and some authors have demonstrated that these phenolic metabolites are more active on inflammatory biomarkers than their precursor structures (parent anthocyanins) [97,98]. In this sense, the bioactivity of anthocyanin metabolites in murta and calafate must continue to be studied in order to achieve adequate information on the biological activity and health-promoting effects derived for the consumption of murta and calafate fruit. 

## 4. Conclusions and Future Perspectives

This review, for the first time, approximates the traditional knowledge of murta and calafate with the scientific research on both species. Scientific knowledge of murta and calafate is much more limited compared to other South American fruits such as maqui (*Aristotelia chilensis* (Mol.) Stuntz) and acai (*Euterpe oleracea* Mart.) for which there are 96 and 232 records (article, proceedings paper, book chapter, and review) available in the Web of Science database, respectively. Advances in the study of the nutritional composition of fruits, the identification of phytochemicals, the validation of traditional use, and the biological activity of certain phytochemicals indicate that murta and calafate are promising sources of natural antioxidants, antimicrobial, and vasodilator compounds with nutraceutical potential. Like international studies on nutraceuticals in other berry-type fruits such as blueberries [95,96,97] future studies are needed to establish the mechanisms of action of both murta and calafate anthocyanins (and their metabolites) in antioxidant and anti-inflammatory activity. These studies are needed in order to further the study of the potential health benefits associated with the consumption of these berry-type fruits. From the nutritional point of view, murta appears to be a good source of ascorbic acid, similar to other fruits in the Myrtaceae family such as camu camu (*Myrciaria dubia* (Kunth) McVaugh, 397 mg/100 g FW) and white guava (*Psidium guajava* L., 142 mg/100 g FW), which are internationally recognized as good sources of vitamin C [99]. 

Plant domestication is a necessary strategy to guarantee the sustainable use of Chilean botanical resources and to standardize the quality of the raw materials derived from murta and calafate. Murta domestication programs are around 20 years old, while calafate has only very recently been domesticated. At the same time, sustainable harvesting of wild murta and calafate performed by several indigenous and rural communities can be an alternative for supplying raw materials for future research. This review will be a useful reference for new research on murta and calafate, respecting and recognizing the traditional knowledge in the hands of indigenous and rural peoples from south and extreme south of Chile. 

## Figures and Tables

**Figure 1 foods-09-00054-f001:**
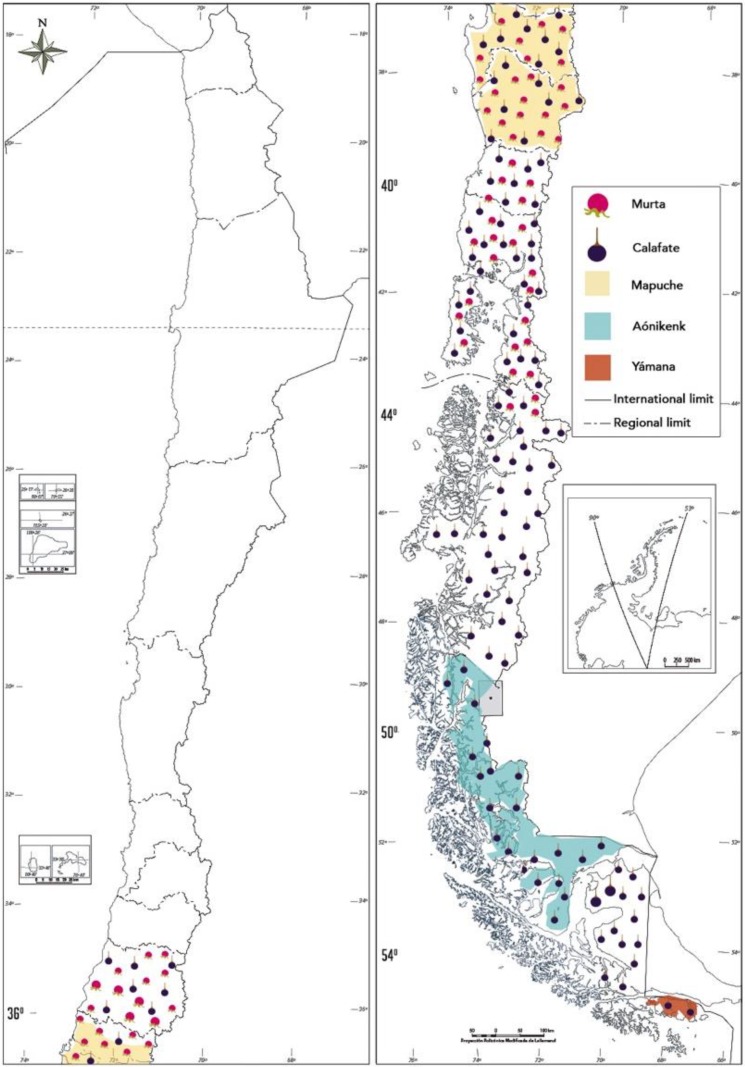
Distribution of murta (**red**) and calafate (**blue**) in a Chilean map, and the territory of Mapuche (**winter white**), Aónikenk (**soft blue**) and Yámana (**brown**).

**Figure 2 foods-09-00054-f002:**
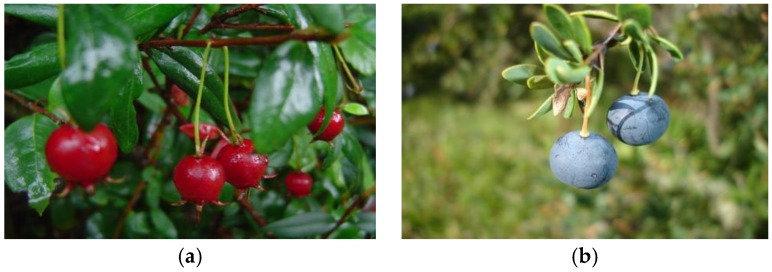
Murta (**a**) and calafate (**b**) fruits.

**Figure 3 foods-09-00054-f003:**
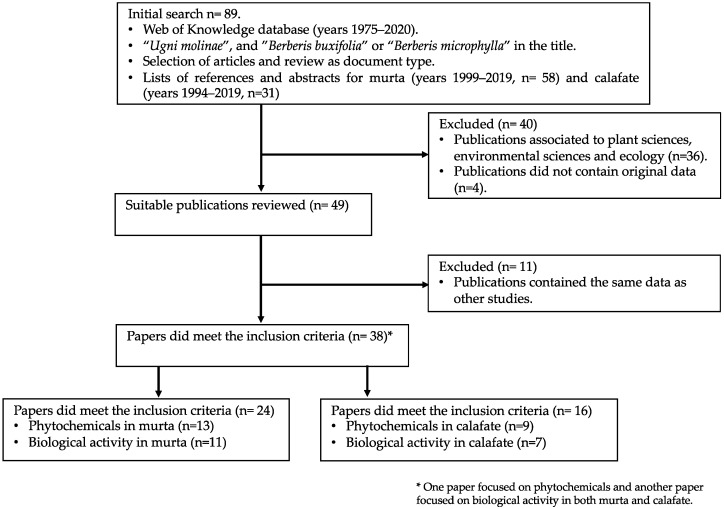
Summary of the search and selection protocols used to identify scientific articles included in the review.

**Figure 4 foods-09-00054-f004:**
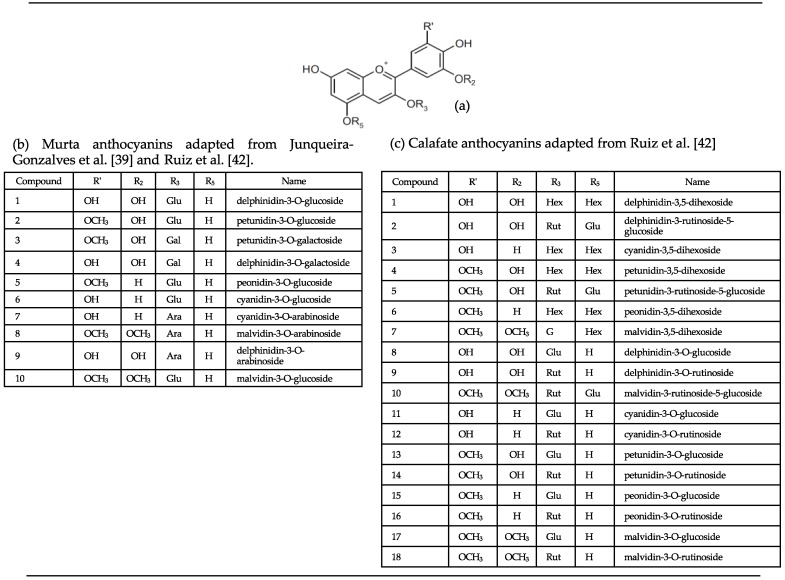
Anthocyanidin basic chemical structure (**a**), anthocyanins in murta (**b**) and calafate (**c**), and their different glycosylation patters.

**Table 1 foods-09-00054-t001:** Traditional medicinal use, biological activity, and phytochemicals reported for murta and calafate.

Plant Organ	Traditional Use	Biological Activity	Phytochemicals	Reference
Murta leaf	Urinary and throat infection	Antioxidant	NR	[66]
		Antioxidant	Phenolic acids, hydrolyzable tannins, epicatechin, myricetin, quercetin	[67]
		Analgesic	Flavonoids and triterpenoids	[68]
		Anti-inflammatory	Triterpenoids	[69]
		Anti-inflammatory	Triterpenoids and phenolic compounds	[70,71]
		Antimicrobial	Catechin, rutin, isoquercitrin, ellagic acid, quercitrin, narcissin, isorhamnetin-3-*O*-glucoside	[72,73,74,75]
Murta fruit	Astringent	Antimicrobial	Isoquinoline alkaloids	[39]
		Vasodilator	Gallic acid, catechin, quercetin-3-*β*-D-glucoside, myricetin, quercetin, and kaempferol	[76,77]
Calafate root	Control fever, anti-inflammatory, stomach pain, indigestion, colitis	Antimicrobial	Isoquinoline alkaloids	[78,79]
		Hypoglycaemic	NR	[80]
Calafate shoot	NR	Antimicrobial	Isoquinoline alkaloids	[78,79,81]
Calafate fruit	NR	Antioxidant	NR	[82]
		Anti-inflammatory	Phenolic compounds	[83]

NR: Not reported.

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
