# Peer review of "Phytochemicals and Traditional Use of Two Southernmost Chilean Berry Fruits: Murta (Ugni molinae Turcz) and Calafate (Berberis buxifolia Lam.)"

_foods, 2020, doi:10.3390/foods9010054_

Round 1

Reviewer 1 Report

Current manuscript provided a comprehensive review on Phytochemicals and traditional use of two 2 southernmost Chilean berry fruits: murta (Ugni 3 molinae Turcz) and calafate (Berberis buxifolia Lam.) It started with the origin and traditional use of murta and calafate among aboriginal people in Chile and then discussed polyphenolics components and their anti-oxidative and anti-inflammatory effects of these two berry fruits. This review manuscript have been nicely written and is fit for Foods journal. There are a few minor revisions that need to be addressed:

1_ Line 63: "This review would like to contribute to the development of new research, indigenous peoples and the 2030 agenda for sustainable development goals of the United Nations". This part is not very clear.

2- "ORAC" needs to be spelled out.

Author Response

Response to reviewer 1

Thank for your comments. Changes in the main manuscript are highlighted in soft blue color.

Line 63: The sentence has been clarified.

Line 278: Oxygen radical absorbance capacity (ORAC) has been included in the abstract.

Reviewer 2 Report

This is a well written  manuscript that documents data on two South American fruits that have not received much attention compared to their 'cousin', Maqui berry!  The authors have carefully chosen the criteria for review and have done a comprehensive job of reviewing. 

Author Response

Thank for your comments.

Reviewer 3 Report

I have revised the manuscript no. 669738

Title: Phytochemicals and traditional use of two southernmost Chilean berry fruits: murta (Ugni molinae Turcz) and calafate (Berberis buxifolia Lam.)

The review aimed to increase the knowledge, under the phytochemical and nutritional aspects, of two berry fruit species only know at the local level (Chile). The review includes many and detailed information about the traditional uses by rural communities of these two species. Moreover, an exhaustive summary of the state of the art regarding their potential human health benefits was reported through bibliographic research.

However, some concerns remain to be clarified

- Row 38-39 family name should be written in font italics.

- Fig. 1 and all references to Fig. 1 (e.g. Fig. 1a), if I read (Fig. 1a) I expect a figure 1 marked with the letter ‘(a)’. Please modify the references to the figures.

- Row 58” warm temperatures…. may favor the plant biosynthesis of anthocyanins’’. The temperatures as also already reported by authors in the following paragraph (row 245) can negatively affect the anthocyanin content. However, commonly the high temperatures have a negative influence on the anthocyanin content. I advise to considering only the effect of temperature oscillation due to low temperatures during the night. The effect of low temperatures is already reported to positively influence the biosynthesis of anthocyanins (Linda Chalker-Scott-1999. Environmental significance of anthocyanins in plant stress responses. Photochemistry and Photobiology, 70, 1-9).

- Row 132 “solid content (SS) of 19 °Brix” when are showed average values is better to use the term ‘’around’’ than ‘’of’’. The same is valid for the other averaged values reported.

-Row 141 “2 g malic acid/100g” can you convert this value in NaOH 100g as reported for murta fruits.

-Row 137 “moisture of 76.9 g/100g” is better reporting this value in %

-Row 201 “the total phenolic compounds in the fruits of wild and selected murta (14-4) genotype were 19.4 and 40.3 mg GAE/g” is fresh or dry weight? Please put the information

-251 “Since the moisture content of fresh fruits is more than 80%” The author stated in row 137 and 145 that the moisture is around 76%. Please be coherent.

-Row 344 “In a comparison of 27 fruit species grown in Chile, the total phenolic content of murta (863 mg GAE/100g FW)” in this paragraph the fruits of murta are associated to very high total phenol content. However, these values, as shown in row 201, 209, and 210, can vary widely. I suggest to rewrite this paragraph more consistently.

-References: all the references need to be revised (e.g. row 11, 12, 13, 14, 16 ecc…)

The authors reported interesting aspects about the nutritional benefits of these two potentials commercially fruits. However, some sentences need to be reviewed and revised. For these reasons, I think that the ms should be minor revised for a publication in (MDPI) Foods.

Author Response

Response to reviewer 3

Thank for your comments. Changes in the main manuscript are highlighted in red.

Family name has been written in italic font (Lines 39-40). Figure 1 and all references to Figure 1 have been clarified for better understanding. Advise of the reviewer has been considered in the text (line 59). Lines 134-135: For mean values in murta fruit, the term “around” has been included. In the same line, for mean values in calafate fruit, the term “around” has been also included (lines 143). Line 144: malic acid has been converted to NaOH as the reviewer suggested. Line 139 and 147: Moisture content has been expressed in % as the reviewer suggested. Line 203: Dry weight has been included as suggested recommended. Line 253: The sentence has been clarified. Line 346. The sentence has been written for better understanding.

10. All the references have been revised.